# Heavy resistance exercise training in older men: A responder and inter-individual variability analysis

Casper Soendenbroe[1,2]*, Jesper L. Andersen[1,2], Mette F. Heisterberg[1], Michael Kjaer[1,2], Abigail L. Mackey[1,2]*

**1** Institute of Sports Medicine Copenhagen, Department of Orthopaedic Surgery, Copenhagen University Hospital - Bispebjerg and Frederiksberg, Copenhagen, Denmark, **2** Department of Clinical Medicine, Faculty of Health and Medical Sciences, University of Copenhagen, Copenhagen, Denmark

\* Caspersoendenbroe@outlook.dk (CS); Abigailmac@sund.ku.dk (ALM)

## Abstract

### Background

The extent of inter-individual variability in response to heavy resistance exercise training (HReT), and the possible existence of non-responders, remains unclear. This study aimed to determine the degree of variability in response to prolonged HReT in healthy older men.

### Methods

We conducted a secondary analysis of an 8- and 16-week intervention involving thrice-weekly HReT (EX) or continuation of a sedentary lifestyle (SED). Fifty-eight healthy men (age 72 ± 5) were randomized to EX (n = 38) or SED (n = 20). Assessments were conducted at baseline, 8-weeks, and 16-weeks for five outcomes: maximal voluntary contraction strength (MVC), rate of force development (RFD), quadriceps cross-sectional area (qCSA), and type I and II myofibre cross-sectional area (fCSA). Inter-individual variability was assessed using the standard deviation of individual responses (SD$_{IR}$). Individual changes relative to a Typical Error were used to classify responders as Poor, Trivial, Robust, or Excellent.

### Results

16 weeks of EX led to group-level increases in MVC (19 ± 14%), RFD (58 ± 80%), qCSA (3 ± 4%), and type II fCSA (14 ± 25%), with no changes in SED. Substantial inter-individual variability was observed. After 16 weeks, 82% of EX participants were classified as Robust or Excellent responders; only 5% were Poor responders. Training compliance and 1RM progression did not explain this variability. Lower baseline

**Data availability statement:** All relevant data are within the paper and its Supporting Information files.

**Funding:** Lundbeck Foundation (R344-2020-254, R402-2022-1387) and Nordea Fonden.

**Competing interests:** The authors have declared that no competing interests exist.

levels were linked to greater improvements but did not fully account for response differences.

## Conclusions

This study provides strong evidence of inter-individual variability in response to HReT among healthy older men. Given the rarity of true non-responders, our data support HReT as the universally recommended first-line strategy for enhancing muscle mass and strength.

---

## 1. Introduction

Heavy resistance exercise training (HReT) is widely recognized as the most effective intervention for increasing muscle strength and muscle mass (i.e., hypertrophy) [1]. Several studies have demonstrated hypertrophy and increases in muscle strength, determined by gold-standard methods (magnetic resonance imaging (MRI) and maximal voluntary contraction strength (MVC)), of 5–15% and 15–30%, respectively, over 2–4 months of training in younger and older individuals [2–5]. However, while these improvements are well-documented at the group level, they mask considerable inter-individual variability. Some individuals experience dramatic strength gains — nearly doubling their initial levels — while others appear largely unresponsive [6,7]. Although several studies have interpreted this response variability as evidence of true inter-individual variability, this assumption is not always supported by rigorous statistical evaluation [8].

In any interventional study, measured changes in a given outcome are influenced by three factors: measurement error, within-person biological variability, and the true effect of the intervention. Accurately identifying true treatment responses requires estimating the magnitude of measurement error and biological variation [9]. This can be achieved by including a non-exercising control group and assuming that random fluctuations and measurement error observed in the control group mirror those in the intervention group [10]. To this end, the standard deviation of individual responses ($SD_{IR}$) quantifies the additional variation observed in the intervention group relative to the control group [10,11]. To the authors' knowledge, only four studies have statistically evaluated response variability following some form of resistance exercise [12–15]. Of these, one focused on a targeted intervention for lower back pain [12], and two involved light-load resistance exercise in individuals with cardiometabolic complications [14,15]. Most notably, Walsh et al., performed a retrospective analysis of a large randomized controlled trial, where adolescents were assigned to HReT, cardiorespiratory training, combined HReT and cardiorespiratory training, or a non-exercising control group for six months. After adjusting for within-person biological variation and measurement error, response variability in both muscle mass and strength outcomes was evident in all exercise groups [13]. Crucially, no studies have yet assessed inter-individual variability following HReT on key outcomes of muscle strength and hypertrophy in healthy older individuals. Given that low muscle mass and strength are linked to physical disability [16], dementia risk [17] and all-cause

mortality [18], understanding variability in HReT response is paramount. As the population aged over 80 has more than tripled in North America and Europe since 1975 and is expected to nearly double again in the next 20 years (United [19]). This makes it more important than ever to develop precise, evidence-based exercise recommendations for older adults.

The primary aim of this study was to determine, using a robust statistical framework, whether inter-individual variability exists in the response to prolonged HReT in healthy older adults. Secondary aims were to classify individual responsiveness to HReT across multiple outcomes and to explore the influence of pre-training levels and training progression on these responses. This study is a secondary analysis of a randomized controlled trial in which healthy older men completed 16 weeks of HReT performed three times per week. To address these aims, we employed a two-pronged approach: first, we assessed the presence of inter-individual variability at a global level; second, we conducted a responder analysis that integrated multiple gold-standard outcomes of muscle mass and strength. Based on the limited number of studies investigating inter-individual variability in response to HReT, we hypothesized that significant inter-individual variability would be present across most outcomes. Additionally, we hypothesized that evaluating changes in multiple outcomes simultaneously would help identify poor and excellent responders to HReT.

## 2. Methods

### 2.1. Study design, and setting

This was a single-center, parallel-group randomized controlled study, approved by The Committees on Health Research Ethics for the Capital Region of Denmark (Reference: H-3-2012-081) and conducted in accordance with the standards set by the Declaration of Helsinki [20]. All participants signed an informed consent form prior to participation. The study was originally designed to evaluate the effect of an angiotensin II type I receptor blocker (losartan) on skeletal muscle adaptations to HReT [21]. There were no effects of receiving losartan on measures of muscle mass and strength, and the two exercise groups are therefore combined into a single exercise group in the present study. A double-blind design was used for medication (losartan or placebo). Participants were led to believe there was a fourth group, receiving placebo with no training.

### 2.2. Study population

Recruitment ran from 10/07/2015–08/03/2016. Males, at or above 64 years of age, were recruited from the greater Copenhagen area. All participants were required to be normotensive, non-smoking and have body mass index (BMI) of 19–34 kg/m². Potential participants also had to be free from major diseases (cancer, organ dysfunctions, ulcers, and liver/kidney/connective tissue diseases), not use blood pressure or anticoagulative medicines. Potential participants had to be sedentary or moderately active, performing no structured strength training or other regular strenuous exercise on a daily or weekly basis, except for activities such as walking or cycling as transportation. This corresponds to Tier 0–1 in the framework proposed by [22].

### 2.3. Randomization

Included participants were block-randomized based on thigh lean mass, their angiotensin-converting enzyme genotype and age, into one of three groups; 1) Losartan + exercise (n = 20), 2) Placebo + exercise (n = 18), or 3) Losartan + continuation of sedentary lifestyle (SED, n = 20). Groups 1 and 2 were combined into a single exercise group in the present study (EX, n = 38). To control for the influence of losartan supplementation, separate analyses was performed for participants receiving losartan (group 1 vs 3; S1 Table).

### 2.4. Intervention

   **2.4.1. HReT.**  The intervention lasted 16 weeks, and participants were tested before (PRE), midway (8wk) and after (16wk). Participants randomised into EX exercised thrice weekly for 16 weeks (48 scheduled sessions). At each session,

participants performed three lower body exercises (seated leg extension, horizontal leg press, and seated leg curl) and two upper body exercises (pulldown and machine shoulder press). For the leg press and leg extension exercises, the training program consisted of six distinct phases that systematically increased training intensity and reduced the number of repetitions.

- Leg press: 3 × 12 at 15 RM (Phase 1), 4 × 10 at 12 RM (Phase 2), 5 × 8–10 at 10 RM (Phase 3), 5 × 6–10 at 8–10 RM (Phase 4), 4 × 6–8 at 8 RM (Phase 5), and 4 × 4–8 at 6–8 RM (Phase 6).

- Leg extension: 3 × 12 at 15 RM (Phase 1), 4 × 10 at 12 RM (Phase 2), 4 × 10 at 10 RM (Phase 3), 5 × 8–10 at 10 RM (Phase 4), 5 × 6–8 at 8 RM (Phase 5), and 4 × 6–8 at 8 RM (Phase 6).

The leg curl followed a similar structured progression but with slightly fewer sets. Further details on the training program are available elsewhere [21]. The 1-repetition maximum (i.e., the heaviest load that can be lifted once) in leg press, leg extension, and leg curl was evaluated before session number 1, 7, 16, 25, 34 and 43. The load used during training was based on the prior 1RM result, although the load was adjusted on a session-by-session basis to secure a high degree of exertion, defined as performing repetitions until concentric failure (inability to complete another repetition with proper technique). All sessions were supervised by study personnel, who also logged weight used and number of repetitions performed for each exercise at each session.

**2.4.2. Medication.** Participants randomized to receive losartan were given a 50 mg losartan pill per day for the first week, and a 100 mg losartan pill per day for the remainder of the study. Participants randomized to receive placebo were given a placebo pill (potato starch, lactose monohydrate, magnesium stearate, gelatine, and talc). The losartan and placebo pills were identical in appearance.

## 2.5. Outcomes

**2.5.1. Maximal voluntary contraction strength and rate of force development.** Using a dynamometer (Kinetic Communicator, model 500 − 11; Chattecx, Chattanooga, TN), isometric maximal voluntary contraction strength (MVC) and peak rate of force development during the initial 200 ms. (RFD) were measured at 70° knee angle (0° equal straight leg). All tests were conducted by the same person. Participants performed three MVC attempts and were instructed to contract "as hard and as fast as possible". MVC and RFD have been published elsewhere as group means [5,21,23].

**2.5.2. Quadriceps cross-sectional area.** Using a Philips Ingenia 3.0 T scanner, both thighs were MRI scanned at the radiology department of Hilleroed Hospital (Copenhagen, Denmark). The cross-sectional area (qCSA) of the quadriceps muscles was manually drawn in OsiriX 8.5 (Pixmeo SARL, Bernex, Switzerland) on the slice closest to 50% femur length. The same person analysed all slices, blinded to group and time. qCSA has been published elsewhere as group means [5,21,23].

**2.5.3. Muscle fiber cross-sectional area.** A total of three muscle biopsies were obtained, using Bergström needles with manual suction [24], from the vastus lateralis muscle of each individual; one at each time point. The samples at PRE and 16wk were taken from the same leg, through different incision sites, 3 cm apart. The sample at 8wk was taken from the contralateral leg. Pieces of muscle tissue were embedded in OCT compound (Tissue-Tek; Sakura Finetek Europe, Alphenaan den Rijn, The Netherlands), and frozen in isopentane (2-Methylbutan; J. T. Baker, Avantor Performance Materials, Deventer, The Netherland) pre-cooled in liquid nitrogen. Samples were stored at −80 °C until further processing. Cross-sections (10 µm) were cut in a cryostat and subjected to ATPase staining at pH 4.37, 4.53, 4.57, and 10.30 to differentiate type I and type II fibers. Stained sections were imaged using a light microscope (Olympus BX40 microscope (Olympus Optical, Tokyo, Japan)), and the borders of individual fibers were manually outlined for calculation of fiber type–specific cross-sectional area (fCSA). The same person analysed all samples, blinded to group and time. fCSA has been published elsewhere as group means [5,21].

**2.5.4. 1-repetition maximum.** 1-repetition maximum (1RM) was evaluated at training sessions 1, 7, 16, 25, 34 and 43 in leg press, leg extension, and leg curl. 1RM has been published elsewhere as group means [21,23].

## 2.6. Data analysis

**2.6.1. Interindividual variability.** To determine the presence of inter-individual variability at the global level in the response to HReT, the SD of individual responses ($SD_{IR}$) was calculated as [10,11]:

$$SD_{IR} = \sqrt{(SD_{EX})^2 - (SD_{SED})^2}$$

Where $SD_{SED}$ and $SD_{EX}$ represent the SD of changes for SED and EX groups, respectively. This $SD_{IR}$ reflects how much the net average treatment effect typically varies between individuals. When inter-individual variability ($SD_{EX} > SE_{SED}$) was present this is indicated by bold-phase. If $SD_{SED}$ is greater than $SD_{EX}$, a negative $SD_{IR}$ value is computed by switching the order of $SD_{SED}$ and $SD_{EX}$ in the equation, which in turn suggests greater variability in the control group [11].

**2.6.2. Individual responses.** Typical errors (TE) were calculated for each outcome as [25]:

$$TE = \frac{(SD_{\Delta SED})}{\sqrt{2}}$$

Where $SD_{\Delta SED}$ represents the SD of changes in SED from test 1–2 and 3 (combined) for each outcome. TE was derived from pooled SED data across both 8- and 16-week intervals. This approach was chosen to ensure a single, consistent threshold for defining individual responsiveness across time points, which we deemed appropriate given the rolling inclusion of participants and overlapping assessments.

Individual change scores (CS) were calculated as:

$$CS = Test\ 2\ or\ 3 - Test\ 1$$

CS of a given outcome that was ≥ the positive TE, was defined as Positive, whereas a change that was ≤ the negative TE was defined as Negative. All changes in between were defined as Neutral.

**2.6.3. Responder classification.** Participants in EX were ranked from 1 (low) to 38 (high) based on their %-change for MVC, RFD, qCSA and type II fCSA. Ranks were then summed across outcomes, adjusted for the number of outcomes in which each participant was represented (ranging from 2−4), and visualized using heatmaps to represent individual responses. To integrate with the individual responses (point 2.6.2), blue and orange markings were used to indicate CS of a given outcome that was greater ≥ or ≤ the TE. Responder classifications were defined as follows: Positive, Neutral, and Negative responses were assigned scores of 1, 0, and −1, respectively. Scores across all outcomes were summed and then divided by the number of outcomes evaluated. Cumulative scores were classified as follows: 0 or below = Poor, 0.01–0.25 = Trivial, 0.26–0.74 = Robust, and 0.75 or above = Excellent.

**2.6.4. Response moderators.** Two potential response moderators were explored. First, it was tested whether progression in training, based on 1RM for leg extension, leg press and leg curl, differed between participants classified as Excellent, Robust, Trivial and Poor, respectively. Second, it was tested whether baseline levels correlated with %-change (PRE to 16wk) for each of the 5 outcomes.

## 2.7. Statistical analysis

Summary data are presented as means ± SD. Baseline participant characteristics were compared between groups using unpaired two-tailed t-tests. Progression in training (1RM) was evaluated using two-way mixed-effects model with session

number (1, 7, 16, 25, 34 and 43) and responder classification (Excellent vs Robust vs Trivial vs Poor) as independent factors, and Tukey's posthoc test. Pearson's correlation was used to explore relationships between baseline levels and %-change values. Data were analysed using Prism (v.10, GraphPad Software and Excel (Microsoft, Redmond, WA, USA). Graphs were prepared in Prism (v.10, GraphPad Software). Statistical significance was set at <0.05.

## 3. Results

### 3.1. Participant characteristics

As shown in Table 1, participants in EX and SED were of similar age, height, weight, and BMI.

### 3.2. Effectiveness of HReT at group level

Participants in EX performed 48 (range 44–53) sessions across 16 weeks (compliance ~95%). As shown in Table 2, MVC, RFD and qCSA was significantly increased in EX by 17.4±12.6%, 42.9±55.8% and 3.6±3.9% at 8 weeks, and by 19.4±14.0%, 58.4±79.5% and 3.4±3.6% at 16 weeks. Type II fCSA was significantly increased in EX by 14.2±24.9%

**Table 1. Participants characteristics.**

| | p (group) | SED (n = 20) | | EX (n = 38) | |
|---|---|---|---|---|---|
| | | Mean ± SD | Range | Mean ± SD | Range |
| Age (yr) | 0.8112 | 72±6 | 66-85 | 72±5 | 65-83 |
| Height (cm) | 0.8941 | 179±7 | 161-190 | 178±7 | 162-191 |
| Weight (kg) | 0.6073 | 83±11 | 62-102 | 85±11 | 57-108 |
| BMI (kg/m²) | 0.4822 | 26±3 | 21-32 | 27±3 | 19-33 |

Data are means±SD with ranges. Abbreviations: SED, Sedentary; EX, Exercise; yr, year.

**Table 2. Group level changes.**

| | Pre to mid | | | | | |
|---|---|---|---|---|---|---|
| | CON | | | EX | | |
| Variable | % | Δ | ES [95%CI] | % | Δ | ES [95%CI] |
| Maximal voluntary contraction [Nm] | 0.17±8.69 | 0.82±14.03 | 0.02 [−0.43 - 0.47] | 17.39±12.56 | 32.65±20.14 | 0.79 [0.42 - 1.17] |
| Rate of force development [Nm/s] | 0.8±34.8 | −65±437 | −0.18 [−0.63 - 0.28] | 42.9±55.8 | 302±329 | 0.82 [0.44 - 1.20] |
| Quadriceps CSA [cm²] | −0.79±2.56 | −0.51±1.52 | −0.04 [−0.48 - 0.39] | 3.56±3.93 | 2.26±2.40 | 0.20 [−0.12 - 0.52] |
| Type I fCSA [µm²] | 3.56±15.17 | 110±653 | 0.12 [−0.34 - 0.59] | 4.03±19.64 | 67±952 | 0.07 [−0.27 - 0.42] |
| Type II fCSA [µm²] | 10.30±20.56 | 277±736 | 0.28 [−0.19 - 0.75] | 4.58±19.20 | 112±827 | 0.11 [−0.23 - 0.46] |
| | Pre to post | | | | | |
| | CON | | | EX | | |
| Variable | % | Δ | ES [95%CI] | % | Δ | ES [95%CI] |
| Maximal voluntary contraction [Nm] | 1.38±7.92 | 3.10±15.03 | 0.08 [−0.38 - 0.53] | 19.36±14.00 | 34.37±22.22 | 0.83 [0.46 - 1.20] |
| Rate of force development [Nm/s] | 8.1±30.3 | 41±345 | 0.11 [−0.34 - 0.56] | 58.4±79.5 | 398±362 | 1.08 [0.68 - 1.48] |
| Quadriceps CSA [cm²] | −0.87±3.27 | −0.62±2.04 | −0.05 [−0.49 - 0.38] | 3.42±3.61 | 2.10±2.25 | 0.18 [−0.14 - 0.51] |
| Type I fCSA [µm²] | −0.96±18.96 | −106±930 | −0.12 [−0.59 - 0.36] | 9.42±19.17 | 385±973 | 0.43 [0.07 - 0.79] |
| Type II fCSA [µm²] | 0.64±24.52 | −60±894 | −0.06 [−0.54 - 0.41] | 14.24±24.90 | 478±867 | 0.49 [0.12 - 0.85] |

Changes at the group level at PRE to 8wk (top) and PRE to 16wk (bottom) shown as percent change, delta values, and effect sizes. Data are means±SD except effect sizes which are show with 95% CI. Abbreviations: SED, Sedentary; EX, Exercise; ES, Effect Size; Nm, newton meter; qCSA, quadriceps cross-sectional area; fCSA, fibre cross-sectional area.

at 16 weeks. No significant group level changes were observed for SED. A direct comparison between exercise groups receiving Losartan or Placebo are provided in S2 Table.

### 3.3. HReT inter-individual variability at the global level

The TE, $SD_{SED}$, $SD_{EX}$, and $SD_{IR}$ can be found in Table 3. Five outcomes were assessed (PRE to 8wk and PRE to 16wk). A positive $SD_{IR}$, indicating greater variability in the response to the intervention in EX compared to SED, was observed in 4/5 outcomes at both 8wk and 16wk. To control for the influence of losartan supplementation we performed the same analyses on only the participants receiving losartan supplementation (excluding the placebo and exercise group). As shown in S1 Table, a positive $SD_{IR}$ was observed in 4/5 and 5/5 outcomes at 8wk and 16wk, respectively.

### 3.4. Individual responses to HReT

Individual CS are shown in Fig 1 for MVC (Fig 1.A), RFD (Fig 1.B), qCSA (Fig 1.C), type I fCSA (Fig 1D) and type II fCSA (Fig 1.E). The TE values used to classify individual responsiveness were: MVC = 10 Nm, RFD = 277 Nm/s, qCSA = 1.25 cm², type I fCSA = 562 μm², and type II fCSA = 582 μm² (Table 3).

**3.4.1. MVC.** For MVC, analyses were performed on 36/38 participants in EX and 19/20 in SED at baseline and 8wk, and on 38/38 and 19/20, respectively, at baseline and 16wk.

For MVC at 8wk, 5 (26%) and 32 (89%) participants in SED and EX respectively, responded positively to the intervention. 2 (11%) participants in SED showed negative responses (Fig 1.A). At 16wk, 6 (32%) and 32 (84%) participants in SED and EX respectively, responded positively to the intervention, with 3 (16%) participants in SED showing a negative response.

**3.4.2. RFD.** For RFD, analyses were performed on 36/38 participants in EX and 19/20 in SED at baseline and 8wk, and on 38/38 and 19/20, respectively, at baseline and 16wk.

For RFD at 8wk, 4 (21%) and 19 (53%) participants in SED and EX respectively, responded positively to the intervention, with 5 (26%) and 1 (3%) participant in SED and EX respectively showing negative responses (Fig 1.B). At 16wk, 5 (26%) and 21 (55%) participants in SED and EX respectively, responded robustly to the intervention, with 3 (16%) and 1 (3%) participant in SED and EX respectively showing a negative response.

**3.4.3. qCSA.** For qCSA, analyses were performed on all participants (38/38 in EX and 20/20 in SED) at baseline, 8wk, and 16wk.

For qCSA at 8wk, 2 (10%) and 24 (63%) participants in SED and EX respectively, responded positively to the intervention, with 7 (35%) and 3 (8%) participants in SED and EX respectively showing negative responses (Fig 1.C). At 16wk, 2

**Table 3. Inter-individual variability at a global level.**

| Variable | TE | Pre to 8wk | | | Pre to 16wk | | |
|---|---|---|---|---|---|---|---|
| | | SDsed | SDex | SDir [95%CI] | SDsed | SDex | SDir [95%CI] |
| Maximal voluntary contraction [Nm] | 10 | 14 | 20 | **14* [−5 - 21]** | 15 | 22 | **16* [−1 - 23]** |
| Rate of force development [Nm/s] | 277 | 437 | 329 | 288* [−227 - 466] | 345 | 362 | **109 [−293 - 331]** |
| Quadriceps CSA [cm²] | 1,3 | 1,5 | 2,4 | **1.86* [−0.67 - 2.54]** | 2 | 2,3 | 0.96 [−1.61 - 2.10] |
| Type I fCSA [μm²] | 562 | 653 | 952 | **693* [−233 - 1008]** | 930 | 973 | **288 [−824 - 919]** |
| Type II fCSA [μm²] | 582 | 736 | 827 | **377 [−597 - 800]** | 894 | 867 | 217 [−788 - 846] |

Key parameters of inter-individual variability for PRE to 8wk and PRE to 16wk. $SD_{IR} = \sqrt{(SD_{EX}^2) - (SD_{SED}^2)}$ with corresponding 95% confidence intervals. Bold-phase indicates when $SD_{EX} > SD_{SED}$, indicating training induced interindividual variability. * indicate when $SD_{IR}$ exceeds the TE. Abbreviations: TE, typical error; $SD_{SED}$, SD of SED; $SD_{EX}$, SD of EX; $SD_{IR}$, SD of individual responses; Nm, newton meter; qCSA, quadriceps cross-sectional area; fCSA, fibre cross-sectional area.

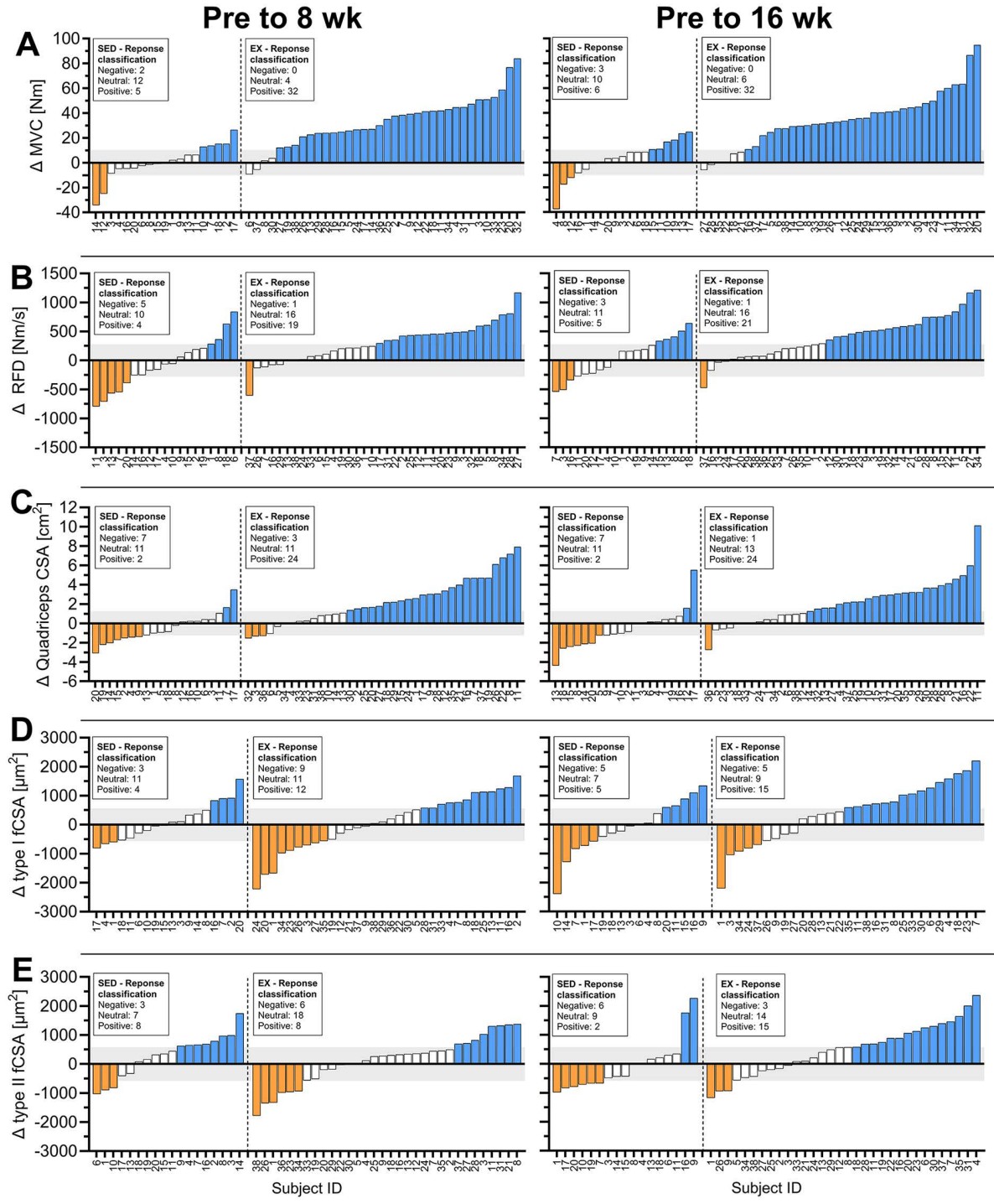

**Fig 1. Individual responses.** Individual change scores (CS) from PRE to 8wk and PRE to 16wk for SED and EX. A) Maximal voluntary contraction (MVC), B) rate of force development (RFD), C) quadriceps cross-sectional area (qCSA), D) type I fibre cross-sectional area (fCSA), E) type II fCSA. Participants are ranked according to their CS, and their responses are categorized as Negative (orange), Neutral (white), or Positive (blue), relative to the typical error (grey zone). Abbreviation: Nm, Newton meter.

(10%) and 24 (63%) participants in SED and EX respectively, responded positively to the intervention, with 7 (35%) and 1 (3%) participant in SED and EX respectively, showing negative responses.

**3.4.4. fCSA.** Analyses were performed on 32/38 participants in EX and 18/20 in SED at baseline and 8wk, and on 32/38 and 17/20, respectively, at baseline and 16wks.

For type I fCSA at 8wk, 4 (22%) and 12 (38%) participants in SED and EX respectively, responded positively to the intervention, with 3 (17%) and 9 (28%) participants in SED and EX respectively, showing negative responses (Fig 1.D). At 16wk, 5 (29%) and 15 (52%) participants in SED and EX respectively, responded positively to the intervention, with 5 (29%) and 5 (17%) participants in SED and EX respectively, showing negative responses.

For type II fCSA at 8wk, 8 (44%) and 8 (25%) participants in SED and EX respectively, responded positively to the intervention, with 3 (17%) and 6 (19%) participants in SED showing negative responses (Fig 1.E). At 16wk, 2 (12%) and 15 (47%) participants in SED and EX respectively, responded positively to the intervention, with 6 (35%) and 3 (9%) participants in SED and EX respectively, showing negative responses.

## 3.5. Responder classification

Heatmaps show the outcome of the rank analyses, with lighter and darker colours indicating percentage change (light = positive, dark = negative). As shown in Fig 2.A, 3 (8%), 9 (24%), 12 (32%) and 14 (37%) participants were defined as Poor, Trivial, Robust and Excellent responders at 8wk. As shown in Fig 2.B, 2 (5%), 5 (13%), 16 (42%) and 15 (39%) participants were defined as Poor, Trivial, Robust and Excellent responders at 16wk.

## 3.6. Response moderators

**3.6.1. Training progression.** As shown in Fig 3.A-C, main effects of session number were observed for 1RM in all exercises (p < 0.0001). Main effects of responder classification were observed for leg press. No interactions between session number (1, 7, 16, 25, 34 and 43) and responder classification was observed for 1RM in any of the three exercises evaluated.

**3.6.2. Baseline levels.** As shown in Fig 4.A-D, baseline levels and changes from baseline to 16wk were significantly correlated for MVC ($R^2$: 0.29, p < 0.0005), RFD ($R^2$: 0.40, p < 0.0001), and type II fCSA ($R^2$: 0.29, p < 0.005).

## 4. Discussion

This study statistically evaluated the presence of inter-individual variability in response to HReT in healthy older men undergoing an 8-week and 16-week intervention using a two-pronged approach. While previous research has assessed individual responses in key outcomes related to muscle strength and/or muscle mass following HReT [6,7,26–42], within-person biological variation and measurement error has not been accounted for [43]. In this study, we found that 8 and 16 weeks of HReT, which significantly increased muscle strength (MVC and RFD) and muscle mass (qCSA and type II fCSA) at the group level, was influenced by inter-individual variability. To explore this further, we classified individual responsiveness using several gold-standard measures of muscle mass and strength, and found that 82% of participants experienced substantial improvements, while only 5% showed limited benefit. Our two-pronged approach was mutually reinforcing – first identifying the presence of inter-individual variability at the global level, then examining which individuals were driving these differences. Collectively, these analyses show that HReT should continue to be universally recommended for healthy older adults. Given the rarity – or possible absence – of true non-responders in this population, future research should prioritize optimizing training strategies broadly and improving the accessibility of HReT, rather than focusing narrowly on the determinants of individual responsiveness. Moreover, future studies should incorporate clinically relevant outcomes such as physical performance, activities of daily living, and quality of life, particularly in frail older adults, to strengthen the translational impact of these findings.

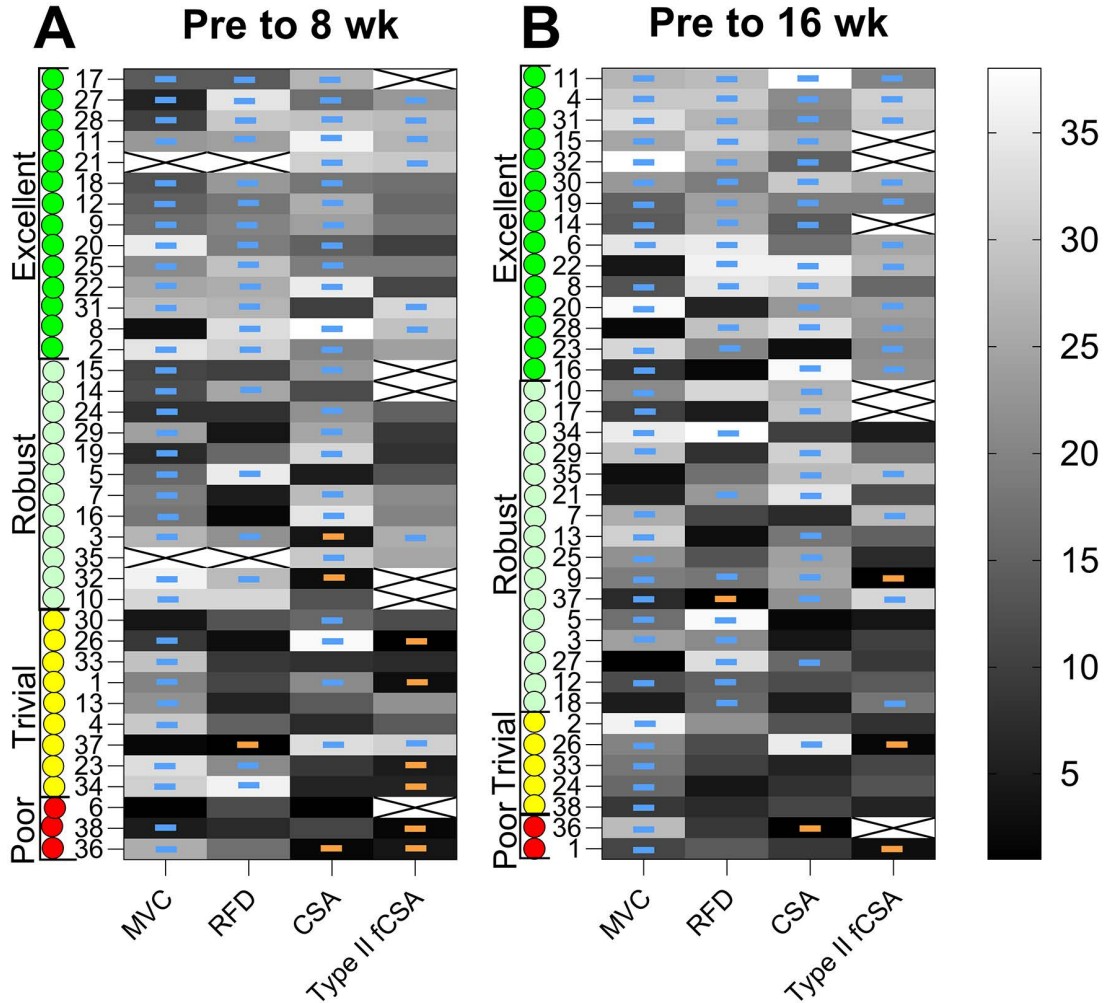

**Fig 2. Responder classification.** Heatmaps showing ranked responses for each outcome, at PRE to 8wk and PRE to 16wk. Blue and orange lines are used to show Positive (≥ the positive TE) and Negative (≤ the negative TE) responses, with all other responses being Neutral (see Fig 1). Based on integration of responses for each outcome, individuals were classified as Excellent (green), Robust (light green), Trivial (Yellow) and Poor (red) responders.

HReT is universally recommended for increasing muscle mass and strength (American College of Sports [44]), and its role in preventing and treating age-related musculoskeletal ailments is acknowledged by researchers, clinicians and practitioners alike [45]. Yet, being unable to mount a substantial response to HReT represents a serious concern, for which there is currently no solution. In 2015, a series of influential papers raised concerns about inappropriately interpreting variability in individual responses to any type of exercise intervention as proof of differences in trainability between individuals [9–11]. In line with these concerns, the present study is the first to demonstrate that inter-individual variability exists at a global level across several gold-standard outcomes of muscle mass and strength following prolonged HReT in older men. Our findings aligns well with the study by Walsh et al., who also observed heterogeneous responses to HReT in adolescents [13]. More broadly, inter-individual variability has been a major focus in the field of exercise physiology, particularly in studies of cardiorespiratory training. Most notably in the HERITAGE Family Study, which revealed a wide range of individual responses to a standardized cardiorespiratory exercise program [46]. However, recent meta-analyses have challenged the assumption that such variability reflects inherent differences in trainability [47,48]. Instead, they

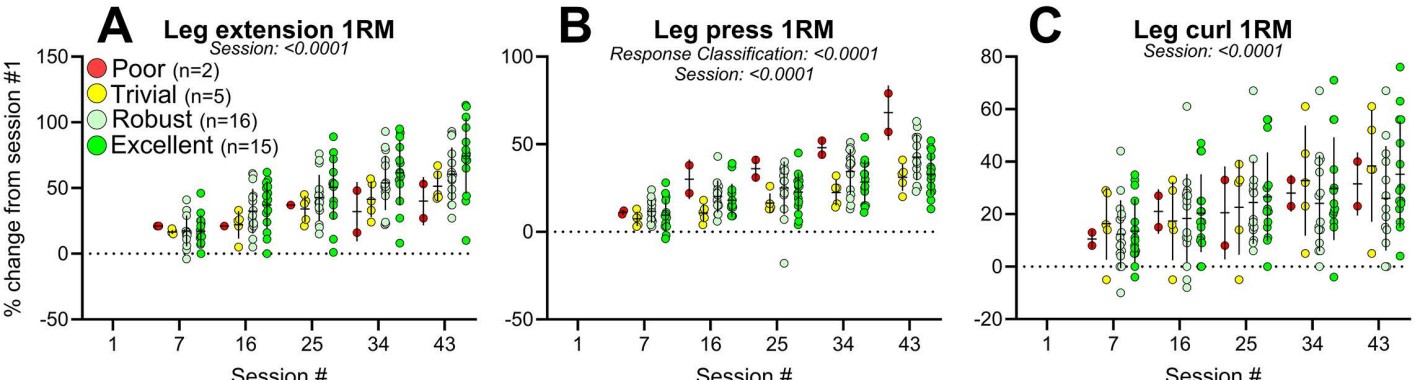

**Fig 3. Training progression.** Percent change in 1RM in leg extension, leg press and leg curl for participants categorized as Poor (red), Trivial (yellow), Robust (light green) and Excellent (green) responders at PRE to 16wk. Data are means±SD with individual data points, and were analysed using two-way mixed-effects model with session number and responder classification as independent factors. Abbreviation: 1RM, 1-repetition maximum.

suggest that much of this variation may be attributed to biological variation or measurement error. Emerging methodological approaches, such as within-subject or contralateral limb designs [49], offer powerful ways to control for systemic sources of variability, though these also come with trade-offs such as potential cross-education effects [50]. These insights highlight that, although the field has progressed considerably since Bouchard and colleagues first brought attention to these issues, the interpretation of individual responses to exercise remains a complex and evolving area of research. It is therefore essential that exercise intervention studies account for sources of variability, typically through the inclusion of a non-exercising control group, when evaluating training-induced adaptations.

Next, we then wanted to examine the proportion of participants that experienced meaningful improvements by stratifying based on whether their change in each outcome exceeded the typical error (TE). Using a threshold such as the TE or the smallest worthwhile change (SWC) lowers the proportion of individuals classified as responders, thereby representing a more conservative approach compared to standard practices [43]. In the present study, only TE was applied as the response threshold; however, alternative thresholds such as SWC could be used in future analyses to explore the robustness of these classifications. By integrating responses across two outcomes of muscle mass (qCSA and type II fCSA) and two outcomes of muscle strength (MVC and RFD), we found that 82% of individuals responded substantially to the intervention. Among the remaining 18%, only two individuals (5%) had limited benefits. Type I fCSA was excluded from this analysis, as no significant group-level change was observed. Notably, the proportion of Poor and Trivial responders declined from 32% at 8 weeks to 18% at 16 weeks. This suggests that a longer training duration may be necessary for some older adults to fully realize the benefits of HReT, highlighting the presence of early and late responders and the importance of maintaining training over time. The finding that two individuals – aged 69 and 72 years – showed limited benefits from 16 weeks of HReT, despite high compliance, raises several important considerations.

Firstly, both participants improved their MVC, yet were classified as poor responders due to a decline in a measure of muscle mass. This naturally prompts the question of the relative importance of strength versus muscle mass. While these two parameters are interrelated, they most often do not change in parallel. Neural adaptations, such as improved motor unit recruitment and firing efficiency, typically manifest earlier, whereas hypertrophic processes begin from the first training sessions but only become detectable later. In this study, MVC increased by approximately 17% at 8wk, whereas this time point was too early to detect a significant increase in type II fCSA. By 16wk, no further improvement in MVC was observed (final increase of 19%), but type II fCSA had increased significantly at the group level (14%). Given that nearly all participants demonstrated early strength gains, responder classification after 8 weeks largely depends on whether

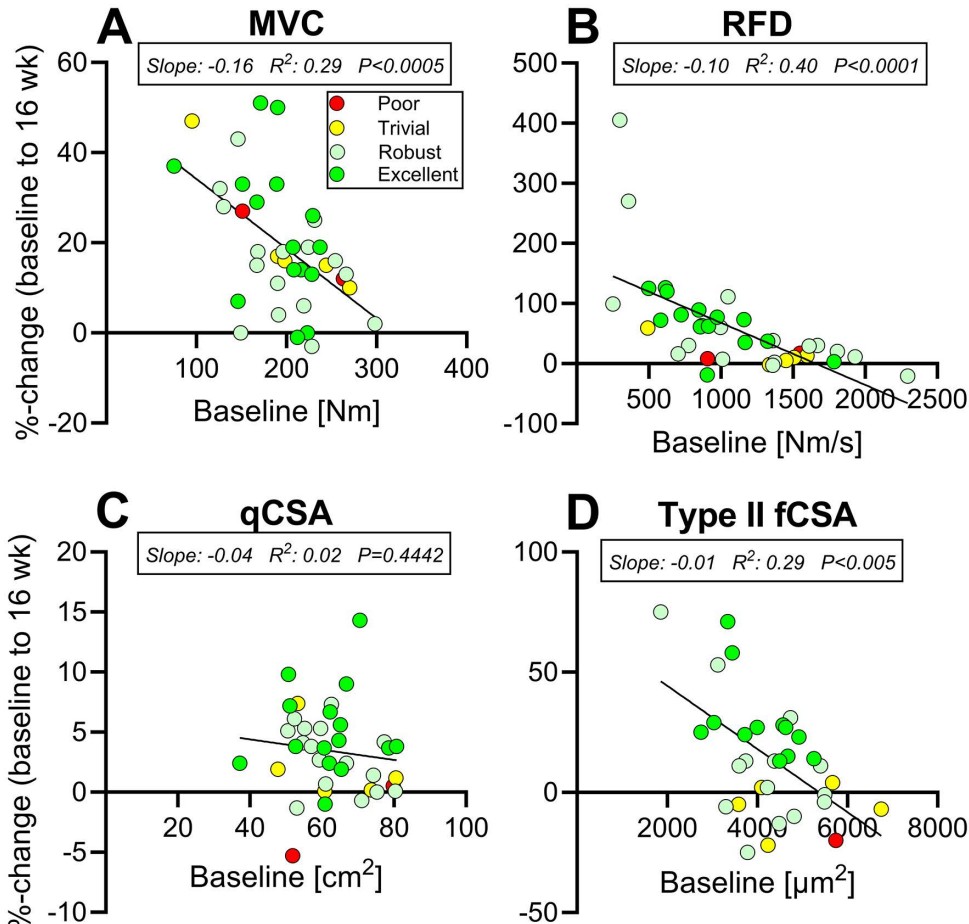

**Fig 4. Baseline levels. Pearson's correlation between baseline levels and percentage change at PRE to 16 weeks for all participants in EX, in MVC (A), RFD (B), qCSA (C), and type II fCSA (D).** Participants are shown according to their responder classification; Poor (red), Trivial (yellow), Robust (light green) and Excellent (green). Linear correlation line is shown, and slope, $R^2$, and P-values were inserted. Abbreviation: MVC, maximal voluntary contraction; RFD, rate of force development; qCSA, quadriceps cross-sectional area; fCSA, fibre cross-sectional area.

these functional improvements were accompanied by measurable hypertrophy, which typically develops on a slower time course. These findings illustrate how technical and biological variation can differ between outcome measures and highlight the challenge of defining responsiveness using a single metric.

Secondly, the finding that only two individuals showed a limited response to HReT highlights the need for a multidimensional framework that considers both structural and functional adaptations. If only muscle mass (qCSA and type II fCSA) had been assessed, 32% of individuals would have been classified as Poor responders. In contrast, assessing only muscle strength would have yielded a 5% Poor responder rate, but with a different set of individuals. This partly explains why studies assessing only one or two outcomes tend to report a higher number of "non-responders," compared to those employing a broader set of outcome measures. For instance, Petrella et al., analyzed fCSA in a population of both young and older men following 16 weeks of HReT and observed 25% of participants to be non-responders [26]. In contrast, Churchward-Venne et al. evaluated multiple outcomes – lean body mass, 1RM (two exercises), fCSA (type I and II), and five-repetition chair-stand performance – in a cohort of older adults and found no non-responders [7].

Thirdly, going back to the present study, it is important to emphasize that that we do not consider the two participants to be "non-responders", and hence we avoid using that term. Both individuals improved their 1RM in all three exercises to a similar or greater extent as those classified as Robust or Excellent responders – indicating that, in practical terms, the training was effective and that 1RM improvements likely reflect meaningful functional gains with clinical relevance. However, we did not include 1RM in our responder classification, as improvements in 1RM are influenced by skill acquisition and neuromuscular efficiency [2], which are less pronounced in assessments of MVC. Including 1RM would therefore confound our results with task-specific learning effects. The observation that improvements in 1RM evaluated every 2–3 weeks do not mirror later gains in muscle mass and strength (as observed at 16wk) suggests that early training progression may not reliably predict long-term adaptations, thereby challenging the practicality of tailoring training programs based on short-term outcomes.

Lastly, we observed that individuals with lower baseline values in MVC (also when normalized to bodyweight), RFD, and type II fCSA - but not qCSA - exhibited greater responses to HReT. While this pattern likely reflects, at least in part, true biological differences in adaptive capacity, it may also be influenced by ceiling effects and the statistical phenomenon of regression to the mean, which can exaggerate the appearance of greater gains among those with lower baseline values. However, baseline values did not explain the overall variability, as Poor and Trivial responders were distributed across the full baseline range. Notably, one Poor responder had high baseline RFD and type II fCSA, potentially limiting their capacity for further improvement. The other Poor responder appeared to lose 5% qCSA over 16 weeks, which seems highly unlikely. As a decline was already evident at 8 weeks, this may suggest an error during the PRE scan and shows that even with a control group not all possible errors can be accounted for. It should also be pointed out that, although prior training history has been suggested to modulate subsequent responsiveness to exercise interventions [51], all participants in the present study were naïve to HReT prior to enrollment. Interestingly, these findings may suggest that individuals with lower baseline muscle strength and mass experience greater relative gains – challenging the common belief that aging muscle is inherently unresponsive to training. In fact, the opposite appears to be true, particularly when age is considered independently of underlying conditions that impair muscle protein metabolism, and when older individuals are appropriately challenged by the training stimulus [3,52].

This study has several limitations that should be considered when interpreting the findings. First, the sample size was modest (n = 38 in the exercise group and n = 20 in the control group), which may limit statistical power to detect subtle effects. Second, the analysis pooled participants who received either losartan or placebo. Although previous work from our group has shown no effect of losartan on muscle maintenance or neural innervation [5,21], we re-calculated $SD_{IR}$ values including only participants who received losartan, and the results were unchanged. Nonetheless, we cannot fully exclude the possibility that subtle effects were not detected. Third, the study did not include a true control group without either exercise or losartan treatment, which may have limited the ability to fully isolate training effects from other influences. Fourth, the study included only healthy older men, and no women or frail individuals were recruited. This homogeneity increases internal validity but limits the generalizability of the findings to broader populations. Fifth, our analytical framework assumes homogeneity of measurement error across groups and time points, an assumption that was not tested. Sixth, the classification of individuals into Poor, Trivial, Robust, and Excellent responders was based on a specific thresholding approach (typical error). While widely used, this approach is inherently arbitrary, and different thresholds or statistical frameworks (e.g., SWC) could yield different classifications. Finally, the present study did not include mechanistic investigations aimed at explaining *why* inter-individual variability occurs. Future work should aim to integrate mechanistic endpoints to help delineate the biological drivers underlying differential training responsiveness.

## 5. Conclusions

HReT is effective at increasing muscle mass and strength at the group level, yet individual responses vary considerably. When multiple outcome domains (muscle mass and strength) and typical error-based thresholds are applied, true

non-responders appear rare among healthy older men. HReT should remain the universally recommended first-line strategy for increasing muscle mass and strength in older adults.

## Supporting information

**S1 Table. Influence of losartan supplementation on global inter-individual variability.** Key parameters of inter-individual variability for PRE to 8wk and PRE to 16wk. $SD_{IR} = \sqrt{(SD_{LOS-EX}^2) - (SD_{LOS-SED}^2)}$. Bold-phase indicates when $SD_{LOS-EX} > SD_{LOS-SED}$, indicating training induced interindividual variability. * Indicate when $SD_{IR}$ exceeds the TE. Abbreviations: TE, typical error; $SD_{LOS-SED}$, SD of losartan with sedentary; $SD_{LOS-EX}$, SD of losartan with exercise; $SD_{IR}$, SD of individual responses; Nm, newton meter; qCSA, quadriceps cross-sectional area; fCSA, fibre cross-sectional area.
(PPTX)

**S2 Table. Group level changes for Exercise groups with Placebo (PLA-EX) or Losartan (LOS-EX).** Changes at the group level at PRE to 8wk (top) and PRE to 16wk (bottom) shown as percent change, delta values, and effect sizes. Data are means ± SD except effect sizes which are show with 95% CI. Abbreviations: PLA-EX, Placebo Exercise; LOS-EX, Losartan Exercise; ES, Effect Size; Nm, newton meter; qCSA, quadriceps cross-sectional area; fCSA, fibre cross-sectional area.
(PPTX)

**S1 Data. Raw data relating to Figs 1, 2 and 4, and Table 2 and 3.**
(XLSX)

**S2 Data. Raw data relating to Fig 3.**
(XLSX)

**S3 Data. Raw data relating to Table 1.**
(XLSX)

## Acknowledgments

The authors are thankful for the technical assistance that was provided by lab technician Camilla Brink Sørensen.

This manuscript was first published as a preprint: Soendenbroe et al., 2025: Heavy Resistance Exercise Training in Older Men: A Responder and Inter-individual Variability Analysis. bioRxiv. https://doi.org/10.1101/2025.05.08.652615

## Author contributions

**Conceptualization:** Casper Soendenbroe, Jesper L. Andersen, Michael Kjaer, Abigail L. Mackey.

**Data curation:** Casper Soendenbroe, Jesper L. Andersen, Abigail L. Mackey.

**Formal analysis:** Casper Soendenbroe, Jesper L. Andersen, Mette F. Heisterberg, Abigail L. Mackey.

**Funding acquisition:** Casper Soendenbroe, Michael Kjaer, Abigail L. Mackey.

**Investigation:** Casper Soendenbroe, Jesper L. Andersen, Mette F. Heisterberg, Michael Kjaer, Abigail L. Mackey.

**Methodology:** Casper Soendenbroe, Jesper L. Andersen, Mette F. Heisterberg, Michael Kjaer, Abigail L. Mackey.

**Project administration:** Casper Soendenbroe, Jesper L. Andersen, Mette F. Heisterberg, Abigail L. Mackey.

**Resources:** Casper Soendenbroe, Jesper L. Andersen, Mette F. Heisterberg, Michael Kjaer, Abigail L. Mackey.

**Software:** Casper Soendenbroe, Jesper L. Andersen, Abigail L. Mackey.

**Supervision:** Casper Soendenbroe, Jesper L. Andersen, Michael Kjaer, Abigail L. Mackey.

**Validation:** Casper Soendenbroe, Jesper L. Andersen, Mette F. Heisterberg, Michael Kjaer, Abigail L. Mackey.

**Visualization:** Casper Soendenbroe, Jesper L. Andersen, Abigail L. Mackey.

**Writing – original draft:** Casper Soendenbroe, Abigail L. Mackey.

**Writing – review & editing:** Casper Soendenbroe, Jesper L. Andersen, Michael Kjaer, Abigail L. Mackey.

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
