## [Decision Letter · Decision Letter 0]

14 Oct 2025

Dear Dr. Soendenbroe,

Thank you for submitting your manuscript to PLOS ONE. After careful consideration, we feel that it has merit but does not fully meet PLOS ONE’s publication criteria as it currently stands. Therefore, we invite you to submit a revised version of the manuscript that addresses the points raised during the review process.

We look forward to receiving your revised manuscript.

Kind regards,

Charlie M. Waugh

Academic Editor

PLOS ONE

Journal Requirements:

“Lundbeck Foundation (R344-2020-254, R402-2022-1387) and Nordea Fonden”

Reviewer's Responses to Questions

**Comments to the Author**

1. Is the manuscript technically sound, and do the data support the conclusions?

Reviewer #1: Yes

Reviewer #2: Yes

Reviewer #3: Yes

2. Has the statistical analysis been performed appropriately and rigorously?

Reviewer #1: No

Reviewer #2: Yes

Reviewer #3: Yes

3. Have the authors made all data underlying the findings in their manuscript fully available?

Reviewer #1: No

Reviewer #2: Yes

Reviewer #3: Yes

4. Is the manuscript presented in an intelligible fashion and written in standard English?

Reviewer #1: Yes

Reviewer #2: Yes

Reviewer #3: Yes

Reviewer #1: The authors found that there is a large degree of inter-individual differences in response to HReT in older males and that non-responders are rare/non-existent. They conclude that HReT should be a universally recommended strategy for improving muscle mass/strength. The authors looked at group effects, as well as individual effects and assessed inter-individual variability by statistically comparing the differences in standard deviations between the control and exercise groups. The research question is novel and is addressed by the methods employed by the authors. Overall, the paper is well-written; however, I have some concerns and comments below to be addressed.

Comments:

• Please run statistics on the participant characteristics and report this in the methods, as well as in Table 1.

• Please add individual datapoints to Figure 3.

• The physical activity of the population recruited is classified as “not performing regular strenuous exercise.” Training status is a well-identified confounder of training response. The authors should provide a more detailed explanation of this inclusion criteria (i.e., how was “strenuous” classified, and how was “regular” classified) so that the initial training status of the volunteers is clearer. The authors should also discuss the degree that differences in initial training status may have impacted the inter-individual differences demonstrated.

• The strength data presented in Figure 4 is presented in absolute terms, rather than normalized to bodyweight or muscle mass. Given that body size largely confounds absolute measures of muscle strength (and bodyweight varied greatly between the participants of this study as the range was reported as 62-102kg), can the authors please comment on why the data were reported in absolute terms? Also, it is mentioned in the discussion that initial strength correlated negatively with strength improvements, which may give the false assumption that the individuals with lower initial strength are less trained; however, we don’t know this for certain, as they may have been the participants of smaller body size. Can the authors please clarify the significance of this correlation when it is being made in absolute terms and explain further why these differences may exist?

• Given that sets, %1RM and reps differed between participants, is it possible that differences in volume accounted for some of the inter-individual differences? If volume can be calculated from exercise logs, this could be a nice addition to the investigation.

• Line 391-393: Please add these calculations to the supplementary file.

• Line 30: for clarity, it would be helpful to the reader if it were clear that the values provided in the abstract are changes after 16 weeks.

• Line 31: value of 82% does not match the value reported in the results section (which was 81%).

• Line 106: not performing? regular strenuous exercise

• Line 122: (15–6)?

• Line 122: Please clarify why the number of sets differed and what determined this difference.

• Line 125: Please clarify how a “high degree of exertion” was indicated.

• Section 2.5.1: Please indicate how many MVC trials were run and how participants were instructed to accurately measure RFD (i.e., were they instructed to “kick as hard and as quickly as possible” during each trial?)

Section 2.5.3

o Please indicate which muscle the biopsy was taken from and the technique used (i.e., Bergström, punch biopsy).

o Please indicate how the muscle was preserved (i.e., flash frozen, embedded in paraffin) and sectioned (i.e., cryostat, microtome).

o Please provide additional details on the fibre-typing stain employed (i.e., incubation times, buffers used).

o Please provide additional details on image acquisition and how the analysis was conducted.

• Line 159: Was a true 1RM measured for each participant, or was a multi-repetition max sometimes used? If a multi-rep max was used, please clarify this in the methods.

• Line 208: Please clarify if significance was accepted at < or ≤ 0.05.

• Table 1: Please define all abbreviations in the table legend.

• Table 2: Please add effect size (ES) abbreviation definition to the table legend.

• Line 241: Please provide an explanation for the missing data demonstrated in Figure 1.

• Line 247: Please provide an explanation for the missing data demonstrated in Figure 1.

• Line 253: Please provide an explanation for the missing data demonstrated in Figure 1.

• Line 259: Please provide an explanation for the missing data demonstrated in Figure 1.

• Line 336: See comment regarding Line 31.

• Line 345-348: Given that neural adaptation to resistance exercise typically precedes structural adaptation (i.e., hypertrophy), it may provide value to comment on this point and the nuance of classifying non/poor-responders (i.e., should strength or muscle size be the key metric? Which is more important for health outcomes?).

• Line 396: “at” the group level.

• Line 457: For clarity of readership, it might be helpful to indicate that the data presented in Figure 4 are from the exercising group only. Stating data is from “all participants” may be misleading.

Reviewer #2: The paper “Heavy Resistance Exercise Training in Older Men: A Responder and Inter-individual Variability Analysis” by Soendenbroe et al. examines the variability in skeletal muscle adaptations among older men following 16 weeks of supervised heavy resistance exercise training (HReT). Fifty-eight healthy men aged approximately 72 years were randomized to an exercise (n = 38) or sedentary control (n = 20) group. Strength and muscle morphology were assessed through maximal voluntary contraction (MVC), rate of force development (RFD), quadriceps cross-sectional area (qCSA), and fibre cross-sectional area (fCSA) of type I and II fibres. Using the standard deviation of individual responses (SDIR) and classification based on changes exceeding the typical error, the authors quantified inter-individual variability and categorized participants as Poor, Trivial, Robust, or Excellent responders.

The study directly addresses a long-standing debate in exercise physiology regarding the prevalence of “non-responders.” By applying a rigorous statistical framework that distinguishes between biological variation and measurement error, and genuine inter-individual differences, the authors advance the field beyond simple descriptive interpretations of variability. The findings support a reassuringly optimistic message for clinicians and policy makers: virtually all older adults benefit meaningfully from structured resistance exercise.

The use of a randomized controlled design, inclusion of a non-exercising comparator, and the integration of multiple gold-standard outcome measures (MRI-derived qCSA, histological fCSA, and dynamometry-based strength) strengthen the validity of the conclusions. The authors’ application of SDIR and typical error–based classification is statistically transparent and aligns with best practices recently advocated in the literature (e.g., Atkinson & Batterham, 2015; Bonafiglia et al., 2021). Moreover, the two-pronged approach, quantifying variability globally and then examining individual trajectories, provides a nuanced understanding that is rarely achieved in training studies.

However, several limitations warrant attention. The sample size, although respectable, limits the power to detect small moderating effects and inflates uncertainty in the estimation of variability. Pooling participants from the losartan and placebo arms of a prior trial could introduce residual confounding, even if sensitivity analyses suggested no drug effect. The study focuses exclusively on older men, which constrains generalizability to women and to frailer or multimorbid populations who may respond differently to loading stimuli. The analysis assumes homogeneity of measurement error between groups and across time points, an assumption that may not hold given the variability of biopsy and MRI results. Furthermore, although the responder classification framework is rigorous, the arbitrary thresholds for categorizing “Robust” versus “Excellent” responders may exaggerate apparent distinctions. It is also notable that the physiological mechanisms underpinning variability, such as neural drive, muscle fibre type distribution, or molecular signalling, were not explored, which limits the ability to explain the observed heterogeneity. Some attention to these issues is warranted.

The discussion effectively contextualizes the findings within the existing literature, contrasting the study with reports of non-responders in both young and older cohorts. The authors’ avoidance of the term “non-responder” is commendable, as most participants improved in at least one outcome. Nevertheless, some claims verge on over-interpretation; concluding that “true non-responders are rare” may not be fully supported, given the modest sample and lack of replication. Additionally, functional outcomes relevant to older adults (e.g., gait speed, chair-rise performance) and quality-of-life-related measures were not included, which would have enhanced clinical translation.

The authors correctly note that baseline strength inversely predicts relative gain, consistent with regression-to-the-mean phenomena. Nonetheless, their framing that HReT should remain a universal prescription is not directly supported by the study’s limited sample and lack of functional outcome measures (e.g., gait speed, balance).

The presentation of individual data (Figures 1–2) is excellent and transparent, aligning with open-science practices. However, the TE derivation relies on pooled SED data across both 8- and 16-week intervals, which may conflate temporal variability. The authors’ responder categorization (Poor, Trivial, Robust, Excellent) is heuristic rather than validated, and no sensitivity analysis using alternative thresholds (e.g., smallest worthwhile change) was shown. Furthermore, conclusions about the rarity of non-responders may be overstated, given the wide confidence intervals for individual effects and potential measurement artifacts (as acknowledged for one apparent case of muscle loss).

Providing confidence intervals for SDIR values in Table 3 could further strengthen the quantitative interpretation.

The discussion could expand on potential mechanistic correlates (e.g., neural vs hypertrophic contributions) to individual variability.

Reviewer #3: This manuscript addresses an important topic in the field of exercise physiology by investigating inter-individual variability in muscle adaptations to resistance training among older adults. The study applies a comprehensive analytical framework and presents individual-level data in a transparent and methodologically grounded manner. Nonetheless, several methodological and interpretative limitations should be considered to contextualize the findings and improve the clarity, robustness, and generalizability of the conclusions.

1. Introduction

Given that this is a secondary analysis, it would be helpful if the authors provided a brief description of the original study design, including sample characteristics, intervention duration, and training protocol. Including this information, particularly in the third paragraph, where the aims and methodological approach are introduced, would improve the clarity and contextualization of the study. Clearly stating the origin and nature of the dataset would also help readers better assess the scope and scientific contribution of the present analysis.

The statement “response variability is widely assumed to reflect true inter-individual variability” may be seen as a rhetorical overgeneralization, since part of the scientific community is already aware of the statistical limitations involved, and many recent studies have applied appropriate analytical approaches (e.g., mixed models, typical error thresholds). A more balanced phrasing is recommended, such as:

“Although several studies assume… this assumption is not always supported by rigorous statistical evaluation.”

2. Methods

2.1. Study Design, and Setting

The authors state that there were no differences between the losartan and placebo groups on outcomes of muscle mass and strength, and therefore merged them into a single exercise group for the current analysis. However, this rationale may be insufficient without reporting the statistical power of the original comparison or the magnitude and precision of the between-group differences (e.g., effect sizes, confidence intervals). A non-significant result does not necessarily imply equivalence, especially if the original study was underpowered to detect meaningful differences. Providing such information would help justify the decision to pool the groups and ensure that the conclusions of the secondary analysis are not biased by an unrecognized pharmacological effect.

2.2. Study Population and 2.3. Randomization

The inclusion and exclusion criteria are well defined, ensuring a relatively homogeneous and healthy older male cohort. The use of stratified block randomization based on physiologically relevant variables (thigh lean mass, ACE genotype, age) strengthens internal validity. However, the exclusive inclusion of males limits the generalizability of findings, and this should be acknowledged in the discussion.

More critically, while the authors combined the two exercise groups (losartan and placebo) for the current analysis, no justification regarding the statistical power of the original comparison is provided. A post hoc equivalence analysis or at least a summary of the between-group results with confidence intervals would be needed to validate this decision. Furthermore, the absence of a true placebo + sedentary group confirms that the study lacked a fully blinded control group without intervention, which should be acknowledged as a methodological limitation.

2.4. Intervention

The resistance training intervention (HReT) is generally well described, with clearly defined duration, frequency, supervised sessions, and progressive intensity based on repeated 1RM testing. These features enhance the internal validity of the study. However, the structure of the six training phases is only briefly mentioned and would benefit from greater detail (e.g., duration, weekly progression, and set/rep schemes) to ensure replicability and allow proper quantification of training volume.

Additionally, no data on training adherence are reported, which are crucial for interpreting inter-individual variability. Differentiating between poor responders and poor compliers requires at least basic adherence metrics (e.g., number of sessions attended). Including this information would strengthen the interpretation of the findings.

2.6. Data Analysis

The outcome measures used in the study are comprehensive and well selected, spanning morphological (qCSA, type II fCSA), functional (MVC, RFD), and histological domains. These were assessed using established gold-standard methodologies, and procedures appear to have been applied with consistency and blinding, which enhances the internal validity of the measurements. Furthermore, the analytical framework employed to characterize interindividual variability is conceptually robust. The authors adopt a widely accepted approach that includes the calculation of SDIR to detect net individual variation beyond random noise, as well as the use of typical error (TE) to classify individual responsiveness. The integration of multiple outcome domains into a composite responder classification provides a broader representation of adaptation and reflects current trends in the field.

However, some critical limitations must be acknowledged. First, the manuscript does not report the measurement reliability parameters, such as TE and coefficient of variation (CV), that are essential to support the interpretation of response classifications. Without these, the accuracy of thresholds used to determine positive, negative, or trivial responses is uncertain. Second, the discretization of individual responses into +1, 0, and –1 categories for each outcome, while practical for visualization, may oversimplify the true biological variability and reduce interpretability. Third, the analysis would be strengthened by the inclusion of formal statistical tests for heterogeneity of variance (e.g., Levene’s test) to complement the descriptive SDIR approach.

Most importantly, a fundamental design limitation restricts the validity of interindividual inferences drawn from the data. The comparison of variability in training response is conducted between two distinct groups (SED vs. EX), each comprising different participants. Such a between-subjects design is inherently vulnerable to confounding factors, including differences in genetic background, biological rhythms, habitual activity, nutrition, and other individual characteristics that may influence the outcomes independently of the intervention. This undermines the ability to isolate true interindividual response variability to the exercise intervention itself. As previously proposed in the literature (Chaves et al., 2025; PMID: 39958513), within-subject designs, where one limb serves as control and the contralateral limb receives the experimental stimulus, provide a more rigorous alternative. These designs inherently control for between-subject biological variation and shared systemic influences such as hormonal fluctuations, sleep, and dietary intake, thereby offering superior sensitivity to detect true variability in responsiveness. It is recommended that the authors explicitly acknowledge, in the Discussion or Limitations section, the implications of employing a between-subject design for interpreting inter-individual variability. A brief mention of alternative approaches, such as within-subject or contralateral limb designs, would enhance the manuscript’s conceptual depth and demonstrate awareness of contemporary methodological advances in the study of individual responsiveness to resistance training.

3. Results

The results section presents individual-level outcomes across multiple domains of muscle adaptation (fCSA, qCSA, MVC, RFD, 1RM), allowing for an integrated view of training responsiveness. The use of individual classification plots and heatmaps is visually effective and aligns with the study’s stated aim of exploring inter-individual variability. The inclusion of a composite responsiveness score adds analytical depth and facilitates the identification of distinct responder subgroups. However, several issues warrant consideration:

First, despite the richness of the dataset, the results are presented in a primarily descriptive fashion. No inferential statistics are used to compare the losartan and placebo groups, nor are formal variance analyses conducted to support the interpretation of inter-individual heterogeneity. This absence weakens the ability to distinguish whether observed differences across participants reflect meaningful biological variability or statistical noise.

Second, although measurements were conducted at three time points (PRE, 8wk, and 16wk), the results focus exclusively on baseline to post-training (PRE–16wk) changes. The omission of temporal trajectories for individual participants is a missed opportunity to explore nonlinear or early adaptive patterns, which could enhance mechanistic understanding.

Third, while the heatmap-based visualization of +1/0/–1 scores allows for intuitive interpretation, the final classification of participants into five responsiveness categories (e.g., overall responders, non-responders, etc.) appears somewhat arbitrary and is not supported by cluster analysis or other multivariate techniques. Furthermore, it is unclear how robust these classifications are to variations in the composite score threshold.

Finally, the decision to pool participants across treatment groups without presenting between-group analyses introduces ambiguity. Although a rationale for this choice is discussed elsewhere, it limits interpretation, especially if subtle treatment effects were present but underpowered to reach significance. It would be helpful to report group-level variability (e.g., SDEX) separately for each treatment arm.

In summary, while the section effectively presents the individual data in a manner consistent with the study’s aims, the lack of inferential comparisons, underutilization of longitudinal data, and limited statistical exploration of clustering or heterogeneity restrict the interpretive strength of the findings.

4. Discussion

The discussion is generally well-written and grounded in the broader scientific literature. The authors clearly articulate the relevance of investigating inter-individual variability in response to RT, particularly in older adults, and contextualize their findings in light of past studies. They appropriately acknowledge prior concerns about misinterpreting individual responses, and adopt a cautious and technically justified approach by incorporating TE thresholds and composite scores. The observation that only two individuals showed limited benefit across four muscle outcomes supports their argument that “true non-responders” may be rare, especially when outcome measures are comprehensive. The authors are also commended for avoiding the term “non-responder” and for discussing the limitations of task-specific improvements such as 1RM. However, a few important considerations should be noted:

First, the interpretation of responders vs. non-responders depends heavily on the reliability of the measurements used. Although the TE approach is methodologically sound, the authors do not report test-retest reliability indices (e.g., TE and, CV) from their own dataset. These are critical to justify the thresholds used for response classification.

Second, the observed discrepancy between gains in MVC and fCSA highlights the limitation of assuming parallel adaptation across structural and functional domains. While the multidimensional framework adopted is commendable, the decision to exclude 1RM due to its learning component could be debated, especially since 1RM improvements may carry significant clinical relevance.

Third, the identification of poor responders based on fCSA or qCSA reductions, particularly in cases where technical error is suspected, underscores the importance of assessing data quality and accounting for sources of error, even when using control groups.

Finally, and importantly, the longitudinal design of the study, with measurements at baseline, 8 weeks, and 16 weeks, provides an excellent opportunity to explore dynamic response trajectories. Although the authors note that the proportion of Poor/Trivial responders decreased from 32% to 18% over time, they do not analyze or discuss whether some participants may be classified as early, late, or sustained responders. This is a missed opportunity. A more detailed analysis of temporal response patterns could shed light on the variability in adaptation kinetics among older adults and offer practical implications for training duration and progression strategies. Classifying individuals by response trajectory, or presenting individual spaghetti plots, would have enriched the interpretation and could inform more personalized exercise interventions.

5. Conclusions

The conclusion is concise and aligns with the main findings of the study. The authors appropriately reiterate that high-load resistance training (HReT) effectively increases muscle mass and strength at the group level and that individual responses show variability. The assertion that true non-responders are rare is supported by their two-pronged analytical approach and the use of multiple outcome domains with TE-based classification. However, the conclusion could be further strengthened by integrating some of the key nuances discussed earlier in the manuscript. Specifically, while the rarity of non-responders is a central takeaway, it is important to emphasize that this conclusion depends on the comprehensiveness of the outcome measures used, the robustness of the measurement protocols, and the choice of statistical thresholds. Additionally, the finding that certain individuals required more time (i.e., responded only by 16 weeks) underscores the importance of training duration and the potential presence of early versus late responders. These temporal aspects are not acknowledged in the final paragraph, despite being critical for tailoring interventions in older populations.

Lastly, the recommendation for HReT as a universally applicable strategy is well justified in light of the data but should be cautiously interpreted in light of participant characteristics (i.e., healthy older men) and potential challenges in generalizing findings to more heterogeneous or frail populations. Including this caveat would enhance the external validity of the concluding statement.

**Do you want your identity to be public for this peer review?** For information about this choice, including consent withdrawal, please see our Privacy Policy

Reviewer #1: No

Reviewer #2: No

Reviewer #3: No

---

## [Author Response · Author response to Decision Letter 1]

31 Oct 2025

Reviewer 1

The authors found that there is a large degree of inter-individual differences in response to HReT in older males and that non-responders are rare/non-existent. They conclude that HReT should be a universally recommended strategy for improving muscle mass/strength. The authors looked at group effects, as well as individual effects and assessed inter-individual variability by statistically comparing the differences in standard deviations between the control and exercise groups. The research question is novel and is addressed by the methods employed by the authors. Overall, the paper is well-written; however, I have some concerns and comments below to be addressed.

We sincerely thank the reviewer for the comprehensive and structured evaluation of our manuscript, as well as for the positive comments regarding its novelty, clarity, and methodological approach. We have carefully considered each of the points raised and revised the manuscript accordingly to address them to the best of our ability.

Comments:

• Please run statistics on the participant characteristics and report this in the methods, as well as in Table 1.

We have now performed statistical comparisons of baseline participant characteristics between groups using independent samples t-tests. The corresponding p-values have been added to Table 1, and the statistical procedure is described in the Methods section: “Baseline participant characteristics were compared between groups using unpaired two-tailed t-tests.”

• Please add individual datapoints to Figure 3.

Individual data points have been added to Figure 3.

• The physical activity of the population recruited is classified as “not performing regular strenuous exercise.” Training status is a well-identified confounder of training response. The authors should provide a more detailed explanation of this inclusion criteria (i.e., how was “strenuous” classified, and how was “regular” classified) so that the initial training status of the volunteers is clearer. The authors should also discuss the degree that differences in initial training status may have impacted the inter-individual differences demonstrated.

We agree that training status is an important determinant of training responsiveness and have clarified the inclusion criteria in the Methods. Participants were classified as non-exercising based on self-reported training history. Specifically, they were sedentary or moderately active, performing no structured strength training or other regular strenuous exercise on a regular basis, except for activities such as walking or cycling as transportation. This corresponds to Tier 0–1 in the framework proposed by McKay et al. (2022) [1].

We have added the following to method: “Potential participants had to be sedentary or moderately active, performing no structured strength training or other regular strenuous exercise on a daily or weekly basis, except for activities such as walking or cycling as transportation. This corresponds to Tier 0–1 in the framework proposed by McKay et al., 2022.”

We have further added a sentence in the discussion to reflect this point: “It should also be pointed out that, although prior training history has been suggested to modulate subsequent responsiveness to exercise interventions [2], all participants in the present study were naïve to HReT prior to enrollment”

• The strength data presented in Figure 4 is presented in absolute terms, rather than normalized to bodyweight or muscle mass. Given that body size largely confounds absolute measures of muscle strength (and bodyweight varied greatly between the participants of this study as the range was reported as 62-102kg), c? Also, it is mentioned in the discussion that initial strength correlated negatively with strength improvements, which may give the false assumption that the individuals with lower initial strength are less trained; however, we don’t know this for certain, as they may have been the participants of smaller body size. Can the authors please clarify the significance of this correlation when it is being made in absolute terms and explain further why these differences may exist?

We agree that normalization of strength data can be relevant in certain contexts, particularly when comparing groups that differ in body size or composition. However, in this study, normalizing to total body weight could introduce additional confounding, as body weight includes fat mass, which is not directly relevant to force production. Given that all participants were older men from a single study cohort and that our primary analyses focused on training-induced changes within individuals, we considered absolute values to be the most appropriate representation of the outcomes. It should also be noted that absolute strength values are commonly reported in similar studies [3–5].

For transparency, we have now provided the data as part of this reply (which will be available online) both in absolute terms and normalized to baseline body weight. As the reviewer rightly notes, body weight does appear to contribute to baseline strength differences; however, the negative association between baseline strength and relative change remains when strength is normalized, indicating that this pattern cannot be explained solely by body size.

To further clarify this point, we have updated the Discussion to read: “Lastly, we observed that individuals with lower baseline values in MVC (also when normalized to bodyweight), RFD, and type II fCSA”

Finally, we emphasize that the negative correlation between baseline strength and training-induced improvements likely reflects a combination of ceiling effects and regression to the mean, rather than differences in body size or training status. This is now explicitly stated in the Discussion: “While this pattern likely reflects, at least in part, true biological differences in adaptive capacity, it may also be influenced by ceiling effects and the statistical phenomenon of regression to the mean, which can exaggerate the appearance of greater gains among those with lower baseline values.”

• Given that sets, %1RM and reps differed between participants, is it possible that differences in volume accounted for some of the inter-individual differences? If volume can be calculated from exercise logs, this could be a nice addition to the investigation.

We appreciate the reviewer’s comment and the opportunity to clarify this point. All participants followed the exact same structured training program and progression, with identical prescribed sets, repetitions, and relative training intensities across all phases. Consequently, there were no systematic differences in training volume between participants. Minor variations may have occurred due to natural fluctuations in load adjustments on a session-to-session basis to maintain high exertion, but these do not represent individualized training volumes.

To make this clearer compared to the previous version, the description of the training intervention has been expanded and now reads: “The intervention lasted 16 weeks, and participants were tested before (PRE), midway (8wk) and after (16wk). Participants randomised into EX exercised thrice weekly for 16 weeks (48 scheduled sessions). At each session, participants performed three lower body exercises (seated leg extension, horizontal leg press, and seated leg curl) and two upper body exercises (pulldown and machine shoulder press). For the leg press and leg extension exercises, the training program consisted of six distinct phases that systematically increased training intensity and reduced the number of repetitions.

- Leg press: 3 × 12 at 15 RM (Phase 1), 4 × 10 at 12 RM (Phase 2), 5 × 8-10 at 10 RM (Phase 3), 5 × 6–10 at 8-10 RM (Phase 4), 4 × 6–8 at 8 RM (Phase 5), and 4 × 4–8 at 6-8 RM (Phase 6).

- Leg extension: 3 × 12 at 15 RM (Phase 1), 4 × 10 at 12 RM (Phase 2), 4 × 10 at 10 RM (Phase 3), 5 × 8–10 at 10 RM (Phase 4), 5 × 6–8 at 8 RM (Phase 5), and 4 × 6–8 at 8 RM (Phase 6).

The leg curl followed a similar structured progression but with slightly fewer sets. Further details on the training program are available elsewhere (Heisterberg et al., 2018). The 1-repetition maximum (i.e. the heaviest load that can be lifted once) in leg press, leg extension, and leg curl was evaluated before session number 1, 7, 16, 25, 34 and 43. The load used during training was based on the prior 1RM result, although the load was adjusted on a session-by-session basis to secure a high degree of exertion, defined as performing repetitions until concentric failure (inability to complete another repetition with proper technique). All sessions were supervised by study personnel, who also logged weight used and number of repetitions performed for each exercise at each session. “

• Line 391-393: Please add these calculations to the supplementary file.

We thank the reviewer for this suggestion. The SDIR calculations were performed using the same statistical procedure already described in the Methods section. Because no additional or unique analytical steps were applied beyond what is already described, we have chosen not to include the raw calculations in the supplementary materials to maintain consistency with how the main analyses are presented.

We do however here provide the calculations as part of the response.

8 week

• MVC: SQRT((19.6^2) - (14.0)) = 14

• RFD: SQRT((437^2) - (365)) = 240 (SED and EX reversed)

• qCSA: SQRT((2.42^2) - (1.52)) = 1.9

• Type I fCSA: SQRT((992^2) - (653)) = 747

• Type II fCSA: SQRT((861^2) - (736)) = 448

16 week

• MVC: SQRT((23.8^2) - (15.0)) = 18

• RFD: SQRT((421^2) - (345)) = 242

• qCSA: SQRT((2.54^2) - (2.0)) = 1.5

• Type I fCSA: SQRT((1062^2) - (930)) = 514

• Type II fCSA: SQRT((942^2) - (894)) = 297

• Line 30: for clarity, it would be helpful to the reader if it were clear that the values provided in the abstract are changes after 16 weeks.

Amended.

• Line 31: value of 82% does not match the value reported in the results section (which was 81%).

The small discrepancy reflects rounding procedures. The proportions at 16 weeks were 42.11% (Robust) and 39.47% (Excellent). When reported separately, each value was rounded down (42% and 39%), whereas their combined value (81.58%) was rounded up to 82%, in accordance with standard rounding rules. We therefore prefer to retain the reported values of 42%, 39%, and 82%, as this is mathematically consistent and does not affect interpretation.

• Line 106: not performing? regular strenuous exercise

Amended.

• Line 122: (15–6)?

• Line 122: Please clarify why the number of sets differed and what determined this difference.

The training program section has been re-written to provide more detail. Importantly, all participants followed the same structured training program, which we have clarified by writing explicitly the number of sets for each of the two main lower body exercises. The revised text now reads: “For the leg press and leg extension exercises, the training program consisted of six distinct phases that systematically increased training intensity and reduced the number of repetitions.

• Leg press: 3 × 12 at 15 RM (Phase 1), 4 × 10 at 12 RM (Phase 2), 5 × 8-10 at 10 RM (Phase 3), 5 × 6–10 at 8-10 RM (Phase 4), 4 × 6–8 at 8 RM (Phase 5), and 4 × 4–8 at 6-8 RM (Phase 6).

• Leg extension: 3 × 12 at 15 RM (Phase 1), 4 × 10 at 12 RM (Phase 2), 4 × 10 at 10 RM (Phase 3), 5 × 8–10 at 10 RM (Phase 4), 5 × 6–8 at 8 RM (Phase 5), and 4 × 6–8 at 8 RM (Phase 6).

The leg curl followed a similar structured progression but with slightly fewer sets. Further details on the training program are available elsewhere.”

• Line 125: Please clarify how a “high degree of exertion” was indicated.

We have clarified this in the text. Specifically, a high degree of exertion refers to performing repetitions until concentric (positive) failure, defined as the inability to complete another repetition with proper technique. The revised sentence now reads: “The load used during training was based on the prior 1RM result, although the load was adjusted on a session-by-session basis to secure a high degree of exertion, defined as performing repetitions until concentric failure (inability to complete another repetition with proper technique).”

• Section 2.5.1: Please indicate how many MVC trials were run and how participants were instructed to accurately measure RFD (i.e., were they instructed to “kick as hard and as quickly as possible” during each trial?)

We have now specified the number of MVC trials and participant instructions in the manuscript. Each participant performed three MVC attempts, and participants were instructed to contract “as hard and as fast as possible”. We have added the following: “Participants performed three MVC attempts and were instructed to contract “as hard and as fast as possible”.

Section 2.5.3

o Please indicate which muscle the biopsy was taken from and the technique used (i.e., Bergström, punch biopsy).

o Please indicate how the muscle was preserved (i.e., flash frozen, embedded in paraffin) and sectioned (i.e., cryostat, microtome).

o Please provide additional details on the fibre-typing stain employed (i.e., incubation times, buffers used).

o Please provide additional details on image acquisition and how the analysis was conducted.

We have expanded the description of the biopsy procedure, sample handling, staining, and analysis to provide the requested detail. The revised text now reads: ” A total of three muscle biopsies were obtained, using Bergström needles with manual suction (Bergstrom, 1975), from the vastus lateralis muscle of each individual; one at each time point. The samples at PRE and 16wk were taken from the same leg, through different incision sites, 3 cm apart. The sample at 8wk was taken from the contralateral leg. Pieces of muscle tissue were embedded in OCT compound (Tissue-Tek; Sakura Finetek Europe, Alphenaan den Rijn, The Netherlands), and frozen in isopentane (2-Methylbutan; J. T. Baker, Avantor Performance Materials, Deventer, The Netherland) pre-cooled in liquid nitrogen. Samples were stored at −80 °C until further processing. Cross-sections (10 µm) were cut in a cryostat and subjected to ATPase staining at pH 4.37, 4.53, 4.57, and 10.30 to differentiate type I and type II fibers. Stained sections were imaged using a light microscope (Olympus BX40 microscope (Olympus Optical, Tokyo, Japan)), and the borders of individual fibers were manually outlined for calculation of fiber type–specific cross-sectional area (fCSA). The same person analysed all samples, blinded to group and time. fCSA has been published elsewhere as group means (Heisterberg et al., 2018; Soendenbroe et al., 2022).”

• Line 159: Was a true 1RM measured for each participant, or was a multi-repetition max sometimes used? If a multi-rep max was used, please clarify this in the methods.

A true 1RM was measured for all participants at all time points.

• Line 208: Please clarify if significance was accepted at < or ≤ 0.05.

< 0.05. Specified.

• Table 1: Please define all abbreviations in the table legend.

Amended.

• Table 2: Please add effect size (ES) abbreviation definition to the table legend.

Amended.

• Line 241: Please provide an explanation for the missing data demonstrated in Figure 1.

• Line 247: Please provide an explanation for the missing data demonstrated in Figure 1.

MVC and RFD were derived from the same test. In the EX group, 8-week data are missing for two participants (21 and 35). For one participant, the data were mistakenly overwritten and thereby lost. For the other, no information is available on why the data are missing. These missing data points are consistent with the dataset reported in previous publications [6,7]. In the SED group, baseline data are missing for one participant, which precluded calculation of Δ and % change values at 8 and 16 weeks. The reason for this missing data is unclear, but this is likewise consistent with previous publications. This has been specified in result section: “For MVC/RFD, analyses were performed on 36/38 participants in EX and 19/20 in SED at baseline and 8wk, and on 38/38 and 19/20, respectively, at baseline and 16wk.”

• Line 253: Please provide an e

---

## [Decision Letter · Decision Letter 1]

28 Nov 2025

Heavy Resistance Exercise Training in Older Men: A Responder and Inter-individual Variability Analysis

PONE-D-25-35642R1

Dear Dr. Soendenbroe,

We’re pleased to inform you that your manuscript has been judged scientifically suitable for publication and will be formally accepted for publication once it meets all outstanding technical requirements.

Kind regards,

Charlie M. Waugh

Academic Editor

PLOS ONE

Additional Editor Comments (optional):

Reviewers' comments:

Reviewer's Responses to Questions

**Comments to the Author**

Reviewer #1: All comments have been addressed

Reviewer #2: All comments have been addressed

Reviewer #3: All comments have been addressed

2. Is the manuscript technically sound, and do the data support the conclusions?

Reviewer #1: Yes

Reviewer #2: Yes

Reviewer #3: Yes

3. Has the statistical analysis been performed appropriately and rigorously?

Reviewer #1: Yes

Reviewer #2: Yes

Reviewer #3: Yes

4. Have the authors made all data underlying the findings in their manuscript fully available?

Reviewer #1: Yes

Reviewer #2: Yes

Reviewer #3: Yes

5. Is the manuscript presented in an intelligible fashion and written in standard English?

Reviewer #1: Yes

Reviewer #2: Yes

Reviewer #3: Yes

Reviewer #1: Thank you for your attention to detail in addressing my concerns, and best of luck with all your future research!

Reviewer #2: The authors have done a nice job. Thank you for addressing the comments. No further action is required.

Reviewer #3: Thank you for the thorough revision. All my previous concerns were fully addressed, and the manuscript is now clear and methodologically sound. I have no further comments.

**Do you want your identity to be public for this peer review?** For information about this choice, including consent withdrawal, please see our Privacy Policy

Reviewer #1: No

Reviewer #2: No

Reviewer #3: No

---

## [Editor Report · Acceptance letter]

PONE-D-25-35642R1

PLOS One

Dear Dr. Soendenbroe,

I'm pleased to inform you that your manuscript has been deemed suitable for publication in PLOS One. Congratulations! Your manuscript is now being handed over to our production team.

Kind regards,

on behalf of

Dr. Charlie M. Waugh

Academic Editor

PLOS One